

# Relationship between hemoglobin glycation index and extent of coronary heart disease in individuals with type 2 diabetes mellitus: a cross-sectional study

Po Chung Cheng[1,*], Shang Ren Hsu[1], Yun Chung Cheng[2,*] and Yu Hsiu Liu[3]

[1] Division of Endocrinology and Metabolism, Department of Internal Medicine, Changhua Christian Hospital, Changhua City, Changhua County, Taiwan

[2] Department of Radiology, Taichung Veterans General Hospital, Taichung, Taiwan

[3] Department of Accounting and Information Systems, National Taichung University of Science and Technology, Taichung, Taiwan

[*] These authors contributed equally to this work.

Corresponding author
Shang Ren Hsu,
wintry_morn@msn.com

## ABSTRACT

**Background**. Individuals with type 2 diabetes (T2D) are at an increased risk of coronary heart disease (CHD). Diabetic complications have recently been associated with a measure of glucose metabolism known as the hemoglobin glycation index (HGI). Currently there is insufficient information regarding a potential link between HGI and cardiovascular disease. This study aimed to investigate the relationship between HGI and extent of CHD in individuals with T2D.

**Methods**. This cross-sectional study screened individuals visiting the endocrinology clinic between June 2012 and May 2016 for eligibility. Enrollment criteria included individuals above 21 years of age with T2D diagnosed in the preceding ten years. Candidates with hemoglobin disorders, pregnancy, and existing coronary artery disease were excluded. Fasting plasma glucose (FPG) and glycated hemoglobin A1c (HbA1c) were sampled three months prior to angiography. The regression equation of predicted $HbA1c = 0.008 \times FPG + 6.28$ described the linear relationship between these variables. HGI was calculated as the difference between the measured HbA1c and predicted HbA1c. Participants were classified into two groups according to the presence of supranormal ($\geq 0$) or subnormal HGI ($< 0$).

**Results**. Among 423 participants, people with supranormal HGI harbored an increased prevalence of multiple vessel disease relative to those with subnormal HGI (Odds ratio (OR): 3.9, 95% CI [2.64–5.98], $P < 0.001$). Moreover, individuals with supranormal HGI more frequently demonstrated lesions involving the left anterior descending artery (OR: 3.0, 95% CI [1.97–4.66], $P < 0.001$). The intergroup difference in mean HbA1c was statistically nonsignificant ($7.5 \pm 1.0\%$ versus $7.4 \pm 1.1\%$, $P = 0.80$).

**Discussion**. This study demonstrated that HGI correlated with the extent of CHD in individuals with T2D. People with supranormal HGI harbored a higher prevalence of extensive cardiovascular disease compared to those with subnormal HGI. The relationship between HGI and extent of CHD enables cardiovascular risk stratification in at risk individuals. Overall, HGI provides useful information concerning cardiovascular risk in clinical practice.

## INTRODUCTION

Type 2 diabetes mellitus (T2D) is a developing epidemic that affects a substantial proportion of the adult population (*Chen, Magliano & Zimmet, 2011*). Changes in dietary habit, urbanization, and sedentary lifestyle contribute to an increasing incidence of the disease (*Yang et al., 2010*). Hyperglycemia exerts detrimental effects on blood vessels, as evidenced by a predisposition to develop retinopathy, nephropathy, and coronary heart disease (CHD) (*Mohammedi et al., 2017*). These vascular complications profoundly influence the quality of life in affected individuals.

Specifically, individuals with T2D are at risk of developing cardiovascular disease (*Shah et al., 2015*), which accounts for nearly sixty percent of diabetes related mortality (*Kalofoutis et al., 2007*). Although glycemic control as represented by glycated hemoglobin A1c (HbA1c) influences vascular disease, this association is not particularly robust (*Laakso, 2010*). Investigators have proposed that elements of hyperglycemia not captured by HbA1c measurement may modify cardiovascular risk (*Fox, 2010*).

The hemoglobin glycation index (HGI) is an indicator of glucose metabolism linked to diabetic complications (*Soros et al., 2010*). HGI correlated with a composite index of cardiac, cerebral, and peripheral vascular events in a recent study involving individuals with T2D (*Nayak et al., 2013*). Specifically, this glycation index may correlate with cardiovascular disease risk. This study investigated the relationship between HGI and extent of coronary vascular disease in people with T2D.

## MATERIALS AND METHODS

### Study population

This cross-sectional study screened patients visiting the endocrinology clinic at Changhua Christian Hospital, Changhua City, Taiwan, between June 2012 and May 2016 for eligibility. Enrollment criteria included individuals exceeding 21 years of age, with T2D diagnosed in the preceding ten years, who received hydroxymethylglutaryl-coenzyme A (HMG-CoA) reductase inhibitors since diabetes onset and underwent coronary angiography (CAG) during the study period. Exclusion criteria involved people who had undertaken CAG prior to the study, those with existing CHD, or who lacked concomitant HbA1c and fasting plasma glucose (FPG) measurements. Candidates with hemoglobin disorders, pregnancy, and congenital coronary artery abnormalities were also ineligible. Decision to perform coronary artery survey was made by cardiologists based on high risk findings on non-invasive testing or high pre-test probability of coronary artery disease. All participants provided written informed consent for CAG. The study was approved by the Institutional Review Board of Changhua Christian Hospital (CCH IRB number: 161111).

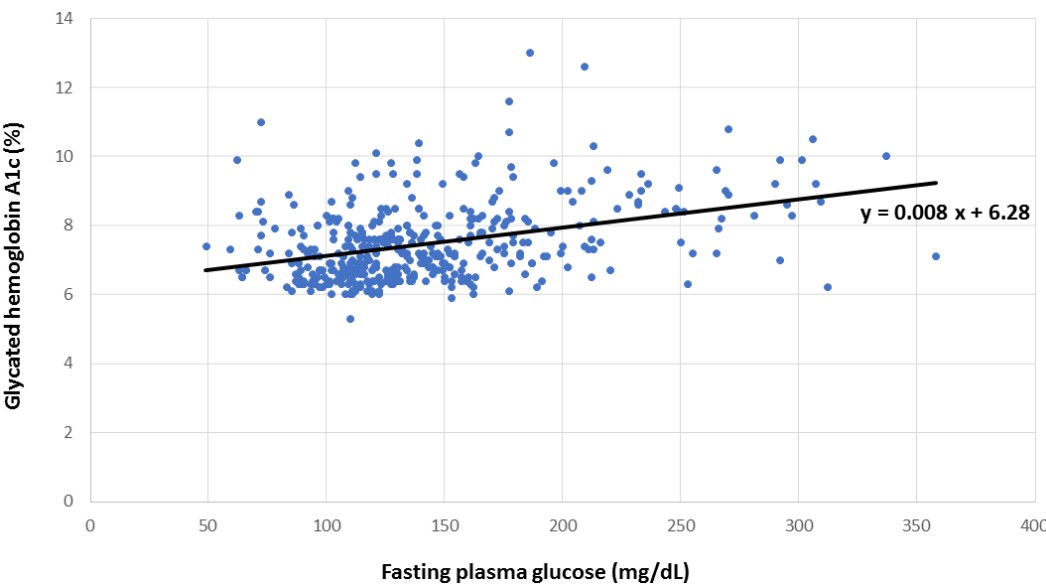

**Figure 1  Relationship between hemoglobin A1c and fasting plasma glucose.** Linear regression for the calculation of predicted hemoglobin A1c from measured hemoglobin A1c and fasting plasma glucose.

## Calculation of HGI

Concomitant HbA1c and FPG measurements from all participants were graphed to establish the linear relationship between these variables, as shown in Fig. 1, from which the regression equation of predicted HbA1c $= 0.008 \times$ FPG $+ 6.28$ was derived. Individual FPG in milligrams per deciliter was substituted into this linear regression equation to derive the predicted HbA1c. HGI was calculated as the difference between the measured HbA1c and predicted HbA1c (*Hempe et al., 2015*). Supranormal HGI was defined as levels above or equivalent to zero, whereas subnormal HGI designated values below zero.

## Classification of CHD

The extent of vascular disease involving the left anterior descending, left circumflex, and right coronary arteries was documented by CAG. Significant stenosis was defined as more than fifty percent narrowing of the diseased vascular segment compared to a proximal or distal normal segment (*Leopold & Faxon, 2015*). Single vessel disease was defined as one or more stenotic lesions in one of the major coronary arteries, whereas multiple vessel disease involved lesions in two or more of the coronary arteries (*Tazaki et al., 2013*). Arteriosclerosis described the observation that none of the stenotic lesions resulted in more than fifty percent narrowing of the major coronary arteries.

## Statistical analysis

Baseline characteristics including age, gender, lipid profile, mean HbA1c, and cigarette smoking were compared between the HGI subgroups. Intergroup comparisons were made using Student's $t$ test for continuous variables and Pearson's $\chi^2$ test for categorical variables. For the HGI subgroups, the prevalence of multiple vessel disease as opposed to single vessel disease or arteriosclerosis was compared using Pearson's $\chi^2$ test. Tests

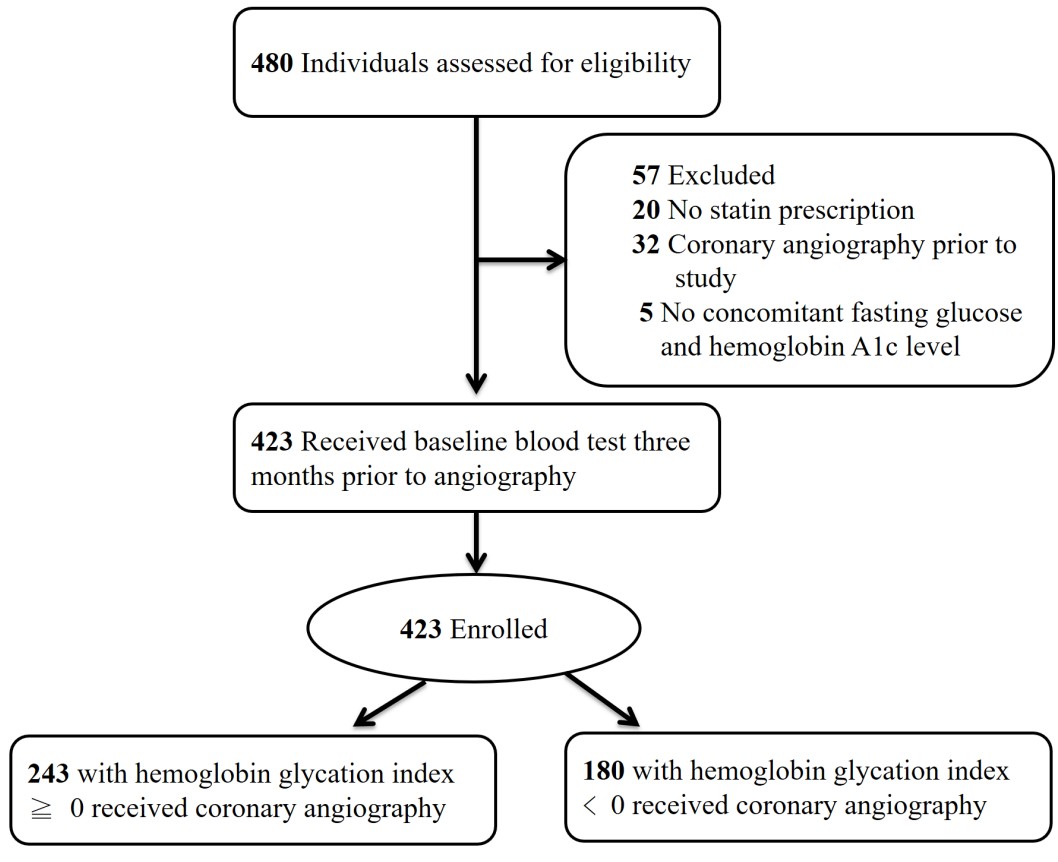

**Figure 2 Enrollment protocol of the study.** Enrollment protocol of the study with number and reason for candidate exclusion.

of statistical significance were based on a two-tailed $P < 0.05$. Statistical analysis was performed using Statistical Package for the Social Sciences (version 22.0, SPSS, Chicago, IL). A multivariate regression analysis was performed to evaluate the contribution of HGI and other risk factors to CHD. A logistic regression analysis adjusted for confounders demonstrated the association between the HGI subgroups and extent of cardiovascular disease.

## RESULTS

Initially 480 individuals with T2D were screened for eligibility. Twenty patients were excluded due to lack of HMG-CoA reductase inhibitor prescription, 32 individuals had received CAG prior to the study and were ineligible, and five candidates were excluded due to absence of concomitantly measured HbA1c and FPG. Figure 2 illustrates the enrollment process of this study.

The study enrolled 423 individuals who were classified according to the presence of either supranormal ($\geq$0) or subnormal HGI (<0). Baseline characteristics including age, gender, and kidney function were similar between groups, as summarized in Table 1. Levels of low-density lipoprotein cholesterol, presence of cigarette smoking, and degree of systolic

**Table 1   Demographic characteristics of the hemoglobin glycation index groups.** Data are expressed as mean with standard deviation for continuous variables and number (%) for categorical variables. Hemoglobin glycation index is defined as the difference between an individual's observed HbA1c and the HbA1c predicted from fasting plasma glucose. HbA1c, glycated hemoglobin A1c; HDL-C, high-density lipoprotein cholesterol; LDLC, low-density lipoprotein cholesterol; TG, triglycerides; SCr, serum creatinine; HGI, hemoglobin glycation index.

|  | HGI < 0 ($n = 180$) | HGI ≥ 0 ($n = 243$) | P value |
|---|---|---|---|
| HbA1c (%) | $7.5 \pm 1.0$ | $7.4 \pm 1.1$ | 0.80 |
| Age (years) | $67 \pm 11$ | $67 \pm 10$ | 0.56 |
| Gender |  |  |  |
| Female | 84 (46.7%) | 98 (40.3%) | 0.19 |
| Male | 96 (53.3%) | 145 (59.7%) |  |
| HDL-C (mg/dL) | $42 \pm 12$ | $39 \pm 14$ | 0.06 |
| LDL-C (mg/dL) | $88 \pm 30$ | $91 \pm 34$ | 0.38 |
| TG (mg/dL) | $159 \pm 126$ | $155 \pm 109$ | 0.76 |
| SCr (mg/dL) | $1.5 \pm 1.7$ | $1.6 \pm 1.7$ | 0.52 |
| Cigarette smoking |  |  |  |
| Yes | 138 (76.7%) | 189 (77.8%) | 0.79 |
| No | 42 (23.3%) | 54 (22.2%) |  |
| Systolic blood pressure (mm Hg) | $136 \pm 24$ | $135 \pm 19$ | 0.65 |
| Proteinuria (g/day) | $0.25 \pm 0.95$ | $0.19 \pm 0.57$ | 0.47 |

**Table 2   Extent of coronary heart disease in hemoglobin glycation index groups.** Data are expressed as number (%) for categorical variables. Hemoglobin glycation index is defined as the difference between an individual's observed HbA1c and the HbA1c predicted from fasting plasma glucose. HGI, hemoglobin glycation index; LAD, left anterior descending artery; CI, confidence interval.

|  | HGI < 0 ($n = 180$) | HGI ≥ 0 ($n = 243$) | Odds ratio | 95% CI | P value |
|---|---|---|---|---|---|
| Arteriosclerosis or single vessel disease | 111 (61.7%) | 70 (28.8%) | 3.9 | 2.64–5.98 | <0.001 |
| Multiple vessel disease | 69 (38.3%) | 173 (71.2%) |  |  |  |
| Left anterior descending artery disease | 102 (56.7%) | 194 (79.8%) | 3.0 | 1.97–4.66 | <0.001 |

blood pressure were also comparable. The intergroup difference in mean HbA1c was nonsignificant ($7.5 \pm 1.0$% versus $7.4 \pm 1.1$%, $P = 0.80$), and both HGI groups harbored similar degree of proteinuria ($0.25 \pm 0.95$ grams per day versus $0.19 \pm 0.57$ grams per day, $P = 0.467$).

As shown in Table 2, individuals with supranormal HGI harbored a higher prevalence of multiple vessel disease relative to those with subnormal HGI (Odds ratio (OR): 3.9, 95% CI [2.64–5.98], $P < 0.001$). Moreover, the supranormal HGI group more frequently demonstrated lesions involving the left anterior descending artery (OR: 3.0, 95% CI [1.97–4.66], $P < 0.001$). As illustrated in Table 3, the length of hospitalization was similar between groups. Intriguingly, people with supranormal HGI demonstrated a trend towards requiring multiple stent deployment relative to those with subnormal HGI (23.0% versus 16.7%, $P = 0.067$).

A multivariate regression analysis was performed using age, systolic blood pressure, low density and high density lipoprotein cholesterol, triglycerides, and HGI to determine the effect of these variables on CHD. The standardized coefficient of each variable in

**Table 3  Clinical outcome of the hemoglobin glycation index groups.** Data are expressed as mean with standard deviation for continuous variables and number (%) for categorical variables. Hemoglobin glycation index is defined as the difference between an individual's observed HbA1c and the HbA1c predicted from fasting plasma glucose. HGI, hemoglobin glycation index.

|  | HGI < 0 (n = 180) | HGI ≥ 0 (n = 243) | P value |
|---|---|---|---|
| Length of stay (days) | 2.8 ± 2.3 | 3.0 ± 2.5 | 0.41 |
| Number of stent |  |  |  |
| None or single | 150 (83.3%) | 187 (77.0%) | 0.067 |
| Multiple | 30 (16.7%) | 56 (23.0%) |  |

**Table 4  Multivariate regression analysis evaluating coronary heart disease as dependent variable.**

| Independent variables | β coefficient | P value |
|---|---|---|
| Age | 0.054 | 0.14 |
| Systolic blood pressure | −0.058 | 0.12 |
| Low density lipoprotein cholesterol | 0.051 | 0.15 |
| High density lipoprotein cholesterol | −0.06 | 0.11 |
| Triglycerides | −0.084 | 0.041 |
| Hemoglobin glycation index | 0.31 | <0.001 |

association with cardiovascular disease is shown in Table 4, which identifies HGI as a prominent contributor to the extent of CHD ($\beta = 0.31$; $P < 0.001$).

A logistic regression analysis also demonstrated the association between HGI and extent of CHD. After adjusting for potential confounding variables, participants with supranormal HGI nonetheless harbored a higher risk of multiple vessel disease (OR: 2.62, 95% CI [1.97–3.49], $P < 0.001$) compared with those with subnormal HGI.

## DISCUSSION

Cardiovascular disease affects a considerable proportion of people with T2D and detracts from their survival (*Naito & Kasai, 2015*; *White et al., 2016*). However, HbA1c measurements delineated only a fraction of cardiovascular disease risk (*Rawshani et al., 2017*). This study demonstrated that HGI consistently correlated with the extent of CHD in T2D. As observed by previous investigators, people with elevated HGI harbor an accelerated rate of protein glycation with subsequent endothelial injury (*Nayak et al., 2011*). Supranormal HGI may also reflect an excess of advanced glycosylation end products that arise from chronic hyperglycemia (*Singh et al., 2014*).

Researchers have postulated that specific patterns of glucose variation, as measured by HGI, constitute a distinctive risk factor for diabetic complications. Whereas HbA1c measures protein glycosylation within the red blood cell, diabetic complications may arise from protein glycation in both the extracellular and intracellular space. HGI was suggested as an indicator of the rate of extracellular protein glycation (*Leslie & Cohen, 2009*). Another potential explanation for a supranormal HGI may be postprandial hyperglycemia, which raises the measured HbA1c above that predicted from FPG (*Rizza, 2010*; *Riddle & Gerstein, 2015*). Indeed, several studies have implicated hyperglycemia after meals in the development

of cardiovascular disease (*Cavalot et al., 2011*; *Node & Inoue, 2009*; *Ceriello, 2009*). These findings were corroborated by the observation in this study that different HGI subgroups, potentially reflecting the degree of postprandial hyperglycemia, consistently correlated with the extent of CHD.

Whereas researchers previously identified an association between HGI and a composite index of cardiac, cerebral, and peripheral vascular events (*Nayak et al., 2013*; *Hempe et al., 2015*), this study provides novel information by focusing on the link between HGI and coronary vascular disease. An implication of this study is that cardiovascular risk assessment may be refined by HGI, particularly in individuals with similar baseline HbA1c. Furthermore, people with supranormal HGI who experience angina pectoris are at risk of CHD and should receive comprehensive examination. Considering the aforementioned link between postprandial hyperglycemia, HGI, and cardiovascular disease risk (*Riddle & Gerstein, 2015*; *Raz et al., 2011*), controlling postprandial hyperglycemia in individuals with supranormal HGI may be an appropriate therapeutic approach to attenuate CHD.

This study benefits from an objective assessment of macrovascular disease by CAG. Conventional risk factors for CHD were similar between the HGI subgroups. Furthermore, potential confounding effects of lipid-lowering therapy were reduced by enrolling recipients of HMG-CoA reductase inhibitors since diabetes outset (*Jellinger et al., 2012*). Data regarding cardiovascular risk factors such as dyslipidemia, blood pressure, and cigarette smoking were available for the entire study population.

Several limitations may arise from the study design. Degree of insulin resistance and mode of antidiabetic treatment may influence CHD but were not uniformly available (*Syed Ikmal et al., 2013*; *Marso et al., 2016*). Although participants received comprehensive diabetes education by certified educators, adherence to lifestyle intervention could not be ascertained. Body weight may influence cardiovascular risk, but investigators previously demonstrated that high-risk coronary anatomy was paradoxically less frequent in obese patients (*Rubinshtein et al., 2006*). The relationship between body weight and CHD remains uncertain and was not assessed in this study. Participants were diagnosed with T2D in the preceding ten years according to their medical records, but precise disease duration could not be confirmed due to the latent nature of T2D. Finally, any complication associated with HGI may be difficult to dissect from the influence of HbA1c (*Sacks, Nathan & Lachin, 2011*).

## CONCLUSIONS

HGI consistently correlated with the extent of CHD in individuals with T2D. People with supranormal HGI harbored a higher prevalence of multiple vessel disease compared to those with subnormal HGI, which may further complicate their management. Undeniably, a correlation between HGI and CHD does not imply causation, and further studies are necessary to explore the physiologic basis of this glycation index. Overall, HGI provides useful information regarding cardiovascular disease risk in T2D.

### Funding

The authors received no funding for this work.

### Competing Interests

The authors declare there are no competing interests.

### Author Contributions

- Po Chung Cheng and Shang Ren Hsu conceived and designed the experiments, performed the experiments, contributed reagents/materials/analysis tools, wrote the paper, reviewed drafts of the paper.
- Yun Chung Cheng analyzed the data, wrote the paper, prepared figures and/or tables, reviewed drafts of the paper.
- Yu Hsiu Liu conceived and designed the experiments, analyzed the data, wrote the paper, prepared figures and/or tables, reviewed drafts of the paper, acts as qualified statistician.

### Human Ethics

The following information was supplied relating to ethical approvals (i.e., approving body and any reference numbers):

The study was approved by the Institutional Review Board of Changhua Christian Hospital (CCH IRB number: 161111).

### Supplemental Information

Supplemental information for this article can be found online at http://dx.doi.org/10.7717/peerj.3875#supplemental-information.

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
