# Peer review of "Relationship between hemoglobin glycation index and extent of coronary heart disease in individuals with type 2 diabetes mellitus: a cross-sectional study"

_PeerJ, doi:10.7717/peerj.3875_

## Round 0.1 · original submission · Major Revisions

· Academic Editor

Major Revisions

Both reviewers recognize that the manuscript has merit, however, some aspects - in particular, methodological issues - need to be further detailed.

Reviewer 1 ·

Basic reporting

Results:
- The authors might consider to create a flowchart to visualize the process of the number of patients screened, number of patients excluded, excluded
- Can the authors show which antidiabetic drug were the patients on?
- The last sentence (line 173) should be deleted from the result section and inserted in the discussion.
- The authors presented a univariate regression analysis for the association between MVD and LAD disease with HGI. The authors need to perform a multivariate regression analysis including well known confounders for CAD severity (e.g. hypertension, hyperlipidemia etc.). Otherwise this is a strong limitation for the study.
- The authors might consider to OR and 95% CI for MVD and LAD disease using fasting glucose or HbA1c alone in comparison to HGI.

Experimental design

Methods:
- Patients had measurement of HGI 3 months before coronary angiography. What were the qualifying reasons for enrollment? How many patients had angina, angina equivalent, SOB (eg. According to CCS and NYHA classification)? What measures were used to schedule an angiography (e.g. positive stress test, high likelihood of CAD estimated by pre-test probability).
- Please clarify, whether patients with existing CAD were excluded or not. If not excluded, add a raw in table 1 showing the percentage of these patients in each HGI category.
- Describe in detail who you calculated the predicted HbA1c.
- Describe the statistical methodology in more detail. Specifically, the regression analysis

Validity of the findings

The findings described by the authors are of particular finding given the fact that HGI easy to calculate and widely available.
However, the authors have to address some issues raised by this reviewer (see below).

Comments for the author

Drs. Cheng et al. performed a nice cross sectional study of patients with DM scheduled for angiography. The authors calculated the HGI before angiography and found an association to CAD severity (MVD and LAQD disease).
The paper is well written and carefully performed. Nevertheless this reviewer has the following comments:
Abstract
- Describe the method for calculation of HGI in the abstract.
Methods:
- Patients had measurement of HGI 3 months before coronary angiography. What were the qualifying reasons for enrollment? How many patients had angina, angina equivalent, SOB (eg. According to CCS and NYHA classification)? What measures were used to schedule an angiography (e.g. positive stress test, high likelihood of CAD estimated by pre-test probability).
- Please clarify, whether patients with existing CAD were excluded or not. If not excluded, add a raw in table 1 showing the percentage of these patients in each HGI category.
- Describe in detail who you calculated the predicted HbA1c.
- Describe the statistical methodology in more detail. Specifically, the regression analysis
Results:
- The authors might consider to create a flowchart to visualize the process of the number of patients screened, number of patients excluded, excluded
- Can the authors show which antidiabetic drug were the patients on?
- The last sentence (line 173) should be deleted from the result section and inserted in the discussion.
- The authors presented a univariate regression analysis for the association between MVD and LAD disease with HGI. The authors need to perform a multivariate regression analysis including well known confounders for CAD severity (e.g. hypertension, hyperlipidemia etc.). Otherwise this is a strong limitation for the study.
- The authors might consider to OR and 95% CI for MVD and LAD disease using fasting glucose or HbA1c alone in comparison to HGI.
Discussion:
- The authors mentioned that HGI reflects an excess of advanced glycosylation end products. Since there is a difference to HbA1c levels, the authors should devote some lines to address these differences. Essentially he relation between HbA1c on one side and advanced glycosylation end products on the other side with CVD severity.
- Since the marker easy to calculate, the authors should describe clinical implications of the study findings.

·

Basic reporting

Clear and unambiguous English is used throughout the text. References are relevant, but most are older than 10 years. More recent references should be used, if available in the literature. Overall, the paper is clear and make simple and significant correlations. Other commentaries may be found in the .pdf document provided.

Experimental design

Experimental design is overall clear and reproducible. Other commentaries may be found in the .pdf document provided.

Validity of the findings

Other commentaries may be found in the .pdf document provided.

Comments for the author

The work is simple and very clear. If reproduced broadly, this correlation between HGI and CHD in T2D patients may provide an efficient tool for diagnosis purposes worldwide. Please refeer to my other commentaries in the .pdf document for further suggestions.

---

## Round 0.2 · Minor Revisions

· Academic Editor

Minor Revisions

Both the reviewers feel that the authors still need to address some issues to improve the manuscript.

Reviewer 1 ·

Basic reporting

No comments.

Experimental design

The study design is accurate. However the description of the estimation of predicted HbA1c is still not clear. The authors need to provide regression coefficients etc. in the graph (figure 1) and explain the where d the numbers in the equation (predicted HbA1c = 0.008 x FPG + 6.28) come from.

Furthermore the reviewers recommendation of insertion of a multivariate OR 95%CI was only commented by the authors and not inserted in the revised version of the paper.

Validity of the findings

The study addresses an important patient population. In such patients developing tools for better patient care are crucial.

Comments for the author

The reviewers comments were only partly addressed in the revised version of the manuscript.
Essentially the authors should avoid discussing their finding in the result section (165-171 of the PDF version of the revised manuscript).
Despite attempts to address the reviewers comment regarding the predicted HbA1c equation, the authors did not provide an understandable explanation in the manuscript. Please provide information where the numbers in the equation come from!

·

Basic reporting

Clear and unambiguous English is used throughout the text. References are relevant, authors removed older references and added more recent ones. Corrections suggested directly into the submission .pdf file were sufficiently fullfiled. All tables are structured accordingly. Supplementary (raw) data is provided separatedly. Despite being clearly presented in Table 2, I believe that there's no need for presenting "yes" or "no" numbers for "LAD disease". By simply indicating "LAD disease" on the left colum and data related to "yes" group (presence of LAD disease) would be enough to imply that the remaining individuals from the groups did not present with the disease. It, however, is not an essential modification, and authors may opt not to make further changes in the refered table. Figures are good and improve undertanding of the data, but I'm not certain about resolution; authors should make sure to provide files with at least 600dpi resolution. If such files were already provided separately, please disregard this recommendation.

Experimental design

Experimental design is overall clear and reproducible. I have no further comments on this section.

Validity of the findings

"no comment"

Comments for the author

Please verify my comments on the "Basic reporting" section. Other minor sugestions are presented at the .pdf file. New recommendations are of minor relevance and authors may or may not accept such changes, considering that they refeer only to the manuscrit, not the study itself. After that, I acept the paper with no further changes.

---

## Round 0.3 · accepted · Accept

· Academic Editor

Accept

Please, before publication, take into account suggestions by Reviewer 2 and myself about small necessary changes in Fig. 2:
bottom panels, please take out the red highlight by "glycation"; move the sign < to the last raw;
second panel from the top, please align "57 excluded".
After these changes, contact the Peer J staff.

Reviewer 1 ·

Basic reporting

The article is presented in high quality.
This reviewer has no further comments

Experimental design

Design is adequate.
This reviewer has no further comments

Validity of the findings

The authors responded to all comments and suggestions made by the reviewers.
This reviewer has no further comments

Comments for the author

Thank you for addressing all comments and suggestion made by the reviewers.
This reviewer has no further comments

·

Basic reporting

There is a highlight mark in the word "glycation" (left and right) from figure 2 - bottom boxes, I believe it's not supposed to be there. Other than that, no comment.

Experimental design

no comment

Validity of the findings

no comment

Comments for the author

no comment